# Pathogenicity of Three *Bursaphelenchus xylophilus* (Steiner & Buhrer) Nickle. Isolates in *Pinus koraiensis* (Siebold & Zucc.) Seedlings

Ye-Fan Cao [1], Lai-Fa Wang [1,*], Xi-Zhuo Wang [1], Xiang Wang [1] and Ming Xu [2]

1   Key Laboratory of Forest Protection of National Forestry and Grassland Administration, Ecology and Nature Conservation Institute, Chinese Academy of Forestry, No. 2 Dongxiaofu, Haidian District, Beijing 100091, China; cyf1995@caf.ac.cn (Y.-F.C.); ladydal@163.com (X.-Z.W.); wx1413@caf.ac.cn (X.W.)
2   Jiangsu Academy of Forestry, No. 109 Danyang Road, Jiangning District, Nanjing 211153, China; xuming2009@126.com
*   Correspondence: nema@caf.ac.cn

**Abstract:** The pine wood nematode (PWN), *Bursaphelenchus xylophilus* (Steiner & Buhrer) Nickle., is one of the most dangerous invasive species in the world, causing devastating pine wilt disease (PWD) in pine trees from many countries. The PWN is now established in 18 provinces in China from the south to north, and it has expanded to some areas of Liaoning Province with temperatures that are beyond the ideal range. It has been reported that *Pinus koraiensis* Siebold & Zucc., one of the representative pine trees of Liaoning Province, has been infected by PWNs. To investigate the pathogenicity of the PWN in *P. koraiensis*, the reproductive ability of PWNs on fungal culture was compared among three isolates: QH-1, NM-1, and CM-1 (QH-1 from Liaoning Province, NM-1 from Nanjing Province, and CM-1 from Chongqing Municipality). Four-year-old *P. koraiensis* seedlings were inoculated with QH-1, NM-1, and CM-1 at a rate of 2000 per seedling. Pathogenicity, external symptoms, and nematode migration were all monitored on a daily basis over the next few days. The results from the experiment showed that all three PWN isolates caused wilt in *P. koraiensis* seedlings, with QH-1 being more virulent than NM-1 and CM-1. In addition, QH-1 exhibited greater reproductive and migration abilities in the seedlings than NM-1 and CM-1. These results indicate that the virulence of the *B. xylophilus* isolates QH-1, NM-1, and CM-1 can differ in terms of seedling mortality, migration ability, and reproductive ability (in trees).

**Keywords:** *Bursaphelenchus xylophilus*; *Pinus koraiensis*; pathogenicity; mortality

## 1. Introduction

The pine wood nematode (PWN), *Bursaphelenchus xylophilus* (Steiner & Buhrer) Nickle., is one of the most dangerous endoparasitic plant nematodes. It causes devastating pine wilt disease (PWD) in pine trees and huge economic and ecological losses in the local pine forests [1,2]. The PWN is native to North America; however, it has now invaded East Asian and European countries, including China, Japan, Korea, Portugal, and Spain [3–7]. The insect vector of PWNs is the *Monochamus* beetle. When *Monochamus* beetles emerge from dead pine trees, they migrate and feed on young, succulent, and healthy pine tree branches. When *Monochamus* beetles feed, they cause wounds and provide entry portals (infection courts) for the PWN [8].

In China, PWD was first detected in 1982 in Nanjing City, Jiangsu Province [9]. With the convenience of modern transportation, the risk of PWD spreading from infected regions to uninfected regions is increasing, and the host species being infected by PWNs are also increasing. In 2013, an investigation showed that *Pinus thunbergii* Parl., *P. densiflora* (Siebold & Zucc.), *P. massoniana* Lamb., *P. thunbergii* × *P. massoniana* hybrid, *P. luchuensis* Mayr, *P. bungeana* (Zucc. ex Endl.), *P. elliottii* Engelm., *P. taeda* (Carl Linnaeus, 1753), and *P. pinaster*

Aiton were susceptible to the PWN [10]. These pine trees are the main hosts of PWNs and are distributed in the areas to the south of the Yangtze River Basin, i.e., the south of China [11]. Due to intentional and accidental human activity, the explosive invasion of PWD has expanded to 18 provinces [12]. Recently, it was found to have spread to the north of China, where the northern pine is threatened [13].

*Pinus. Koraiensis* (Siebold & Zucc.), generally called Korean pine, is native to Korea, China, eastern Russia, and Japan [14], where it is used as an ornamental and economic tree species. The pine nuts of *P. koraiensis* are well known to be a source of fatty acids, mostly trienoic acid and pinolenic acid [15]. *P. koraiensis* is also a source of turpentine resin and tannin [16]. In China, *P. koraiensis* is abundant in the northeast regions and is distributed throughout Manchuria. In 2017, *P. koraiensis* was first reported to be infected by the PWN in Fengcheng, Liaoning Province, China [17]. The PWN vector was also found in the same region of the *P. koraiensis* forests infected by PWNs [18]. Thus, PWD has invaded the northeast of China, and the security of *P. koraiensis* forests is in danger.

Previous studies have shown that different factors induce alterations in the pathogenicity of the PWN in pine trees [19–23]. According to some studies, there is a positive correlation between pathogenicity and migration; PWN isolates with faster migration have higher pathogenicity [24], and virulent PWN isolates cannot migrate easily at low temperatures. Many others have discovered no significant relationship between pathogenicity and migration [25] and have alleged population buildup as a key factor in the pathogenicity of the PWN. Because highly virulent PWNs have a rapid and high reproductive capacity, they can quickly build up a population and cause pine trees to wilt [26,27]. Moreover, virulent isolates and avirulent isolates have been shown to respond differently to susceptible pines. In one study, virulent isolates were more tolerant to reactive oxygen species (ROS) produced by the host than avirulent isolates [28]. In this study, we investigated the factors that could have a relationship with the pathogenicity of the PWN in *P. koraiensis* via inoculation with three PWN isolates. The aim of this experiment was to (i) determine differences in pathogenicity among the three isolates and (ii) investigate the relationships between the different factors and the PWN.

## 2. Materials and Methods

### 2.1. Nematodes

In this study, we used three different PWN isolates (NM-1, CM-1, and QH-1) for inoculation. NM-1 and CM-1 were extracted from infected *P. thunbergii* in Jiangsu and Chongqing, China. QH-1 was extracted from infected *P. koraiensis* in Liaoning, China [29]. All nematodes were cultured on *Botrytis cinerea* Pers. (1794) fungus grown on barley grains at 25 °C for two weeks. The nematodes were extracted using the Baermann funnel technique at room temperature [30]. The suspension of nematodes was adjusted to a concentration of 10,000 nematodes per ml with sterile water.

### 2.2. Plant Materials

Four-year-old *P. koraiensis* seedlings were grown in a nursery located at the Ecology and Nature Conservation Institute in Beijing, China. The seedlings were transplanted in plastic pots and moved to the nursery in the spring of 2019. To avoid the influence of abiotic stress factors, the experimental seedlings were domesticated for a few months. Unhealthy seedlings (needle yellowing or wilting) were thrown out during the domestication period.

### 2.3. Reproductive Ability of B. xylophilus

*B. cinerea* was cultured on potato dextrose agar (PDA) medium in a Petri dish with a diameter of 9.0 cm. The dish was placed at 25 °C in the dark for 2 weeks. Each *B. xylophilus* isolate was multiplied on a Petri dish of *B. cinerea* in PDA medium and incubated for 2 weeks at 25 °C in the dark. Following incubation, the nematodes were extracted from the dish using the Baermann funnel method at room temperature. The number of nematodes

in the funnel was counted using a stereomicroscope. Ten replications were conducted for each isolate.

### 2.4. Pathogenicity Test

Pathogenicity tests for *B. xylophilus* were carried out during the summer months of 2019 (from June to August) in a nursery located at the Ecology and Nature Conservation Institute in Beijing, China. Ten Korean pine seedlings were inoculated with a 200 μL suspension of the three isolates of *B. xylophilus* containing 2000 nematodes from each isolate. The inoculation point was about 2 cm under the middle of the seedlings. The wound for inoculation was cut with a 22-gauge scalpel. As a negative control, ten seedlings of each pine species were inoculated with sterile water. After the inoculation, the development of pine wilt disease was observed. Wilt symptoms, including leaf discoloration and shoot wilting, were observed and recorded every day for two months. After 35 days, the mortality rate of the inoculated seedlings was determined, and *B. xylophilus* was extracted from the diseased or dead seedlings for counting.

### 2.5. Migration of B. xylophilus in Inoculated Trees

To observe the migration of the different isolates of *B. xylophilus* in *P. koraiensis*, 24 Korean pine seedlings were inoculated with NM-1, CM-1, and QH-1 nematodes during the summer months of 2019 (from June to July). Each seedling was inoculated with a 200 μL suspension of *B. xylophilus*, which contained 2000 nematodes. All seedlings were cut into 1 cm segments from the inoculation point (1.5 cm) in both upward and downward directions on days 1, 2, 4, 7, 10, 15, 20, and 25 after inoculation. Nematodes were extracted from the cut segments using the Baermann funnel method and counted using a stereomicroscope.

### 2.6. Internal Symptom Observations in Inoculated Trees

Observations of the internal symptoms in diseased or dead inoculated seedlings were carried out during the summer months of 2019 (from June to July). Eighteen *P. koraiensis* seedlings were inoculated with NM-1, CM-1, and QH-1. Each seedling was inoculated with 2000 nematodes. Wood segments (1 cm long) were cut from the inoculation point in a downward direction at 1, 2, 4, 7, 10, and 25 days after inoculation. Sections of the wood segments were observed with a light microscope.

### 2.7. Data Analysis

SPSS (13.0 for Windows) was used to analyze the data. The mortality rate of diseased pines was calculated as the percentage of the sum of dead to partly dead seedlings 35 days after inoculation. The data were analyzed using an analysis of variance (ANOVA), and Tukey tests were used to separate the means. The number of nematodes was presented as the mean $\pm$ SE. A migration thermodynamic diagram of *B. xylophilus* was generated using Origin 2017 (for Windows) and SigmaPlot 14.0 (for Windows).

## 3. Results

### 3.1. Reproductive Ability of B. xylophilus

The mean numbers of reproducing nematodes from the different isolates of *B. cinerea* are shown in Figure 1. The average numbers of CM-1, QH-1, and NM-1 on the fungal mats were 18,559, 20,706, and 24,870, respectively. The mean number of NM-1 *B. xylophilus* isolates was significantly larger than that of CM-1 and QH-1 ($p < 0.01$).

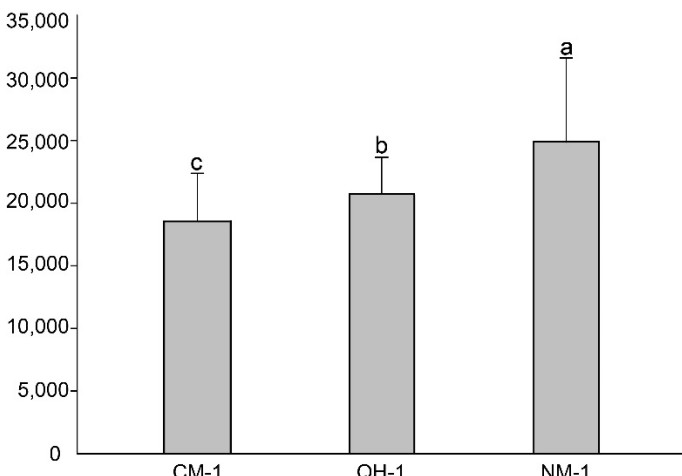

**Figure 1.** Reproductive ability of *Bursaphelenchus xylophilus* (Steiner & Buhrer) Nickle.on *Botrytis cinerea* Pers. (1794) over a period of 2 weeks. Different letters above the columns of values indicate significant differences ($p < 0.01$).

### 3.2. Disease Development in P. koraiensis Seedlings

The early symptoms of disease in the inoculated *P. koraiensis* seedlings were needle discoloration and wilting (Figure 2). Needle discoloration was observed near the inoculation point 10 days after inoculation with isolate QH-1, while isolates NM-1 and CM-1 caused discoloration 15 days after inoculation. No disease symptoms occurred in the seedlings treated with sterile water. The intermediate symptoms of the inoculated seedlings were increased wilting and discoloration. One-third of the leaves from the *P. koraiensis* seedlings inoculated with QH-1 were discolored 15 days after inoculation. Five weeks after inoculation, the mortality rates of *P. koraiensis* seedlings inoculated with QH-1, NM-1, and CM-1 were 100%, 60%, and 40%, respectively. No mortality occurred among the seedlings inoculated with sterile water.

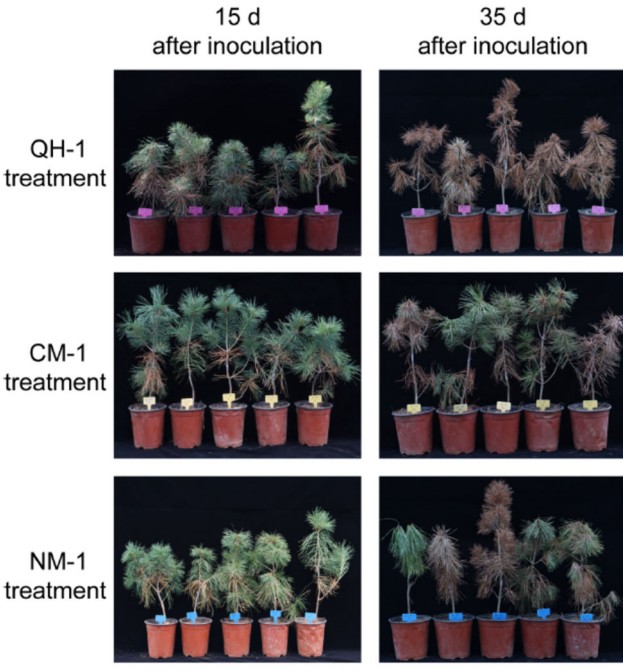

**Figure 2.** Development of wilt symptoms in *Pinus. Koraiensis* (Siebold & Zucc.) seedlings inoculated with *B. xylophilus*.

The mean number of *B. xylophilus* differed significantly between the *P. koraiensis* seedlings inoculated with different isolates ($p < 0.01$) (Table 1). The mean number of nematodes extracted from diseased or dead *P. koraiensis* seedlings inoculated with QH-1 was greater than that of the seedlings inoculated with NM-1 and CM-1 (Table 1). No nematodes were collected from the seedlings inoculated with sterile water.

**Table 1.** Mortality of seedlings inoculated with *Bursaphelenchus xylophilus* (Steiner & Buhrer) Nickle. and nematode population densities.

| Nematode Isolate | Seedling Species | Number of Seedlings | Mortality Rate (%) | Nematodes in Seedlings [1] |
|---|---|---|---|---|
| QH-1 | 4-year-old | 10 | 100 | 28,776 ± 4774 [a] |
| NM-1 | seedlings | 10 | 60 | 19,290 ± 3502 [b] |
| CM-1 | *P. koraiensis* | 10 | 40 | 19,737 ± 3501 [b] |
| CK | (Siebold & Zucc.) | 10 | 0 | 0 |

Note: [1] recovery of nematodes was carried out 5 weeks after inoculation. Values represent the mean ± SD. Different letters represent significantly different values according to a *t*-test ($p < 0.01$).

### 3.3. Distribution of B. xylophilus in Inoculated Trees

The distributions of the different isolates of *B. xylophilus* in inoculated *P. koraiensis* seedlings are shown in Figure 3. Nematodes were found in the stems near the inoculation point (1–2 cm) 2 days after inoculation; however, the number of nematodes was extremely small, and the migrating distance of QH-1 was greater than that of CM-1 and NJ-1. Ten days after inoculation, the number of nematode populations in the QH-1 treatment group was significantly higher than that of the CM-1 and NJ-1 treatment groups (Figure 4). The migrating distance of QH-1 was 8 cm (from the inoculation point), while that of CM-1 and NM-1 was much lower. Over the next few days, significant differences were found between the numbers of *B. xylophilus* populations among the three different isolates ($p < 0.01$), similar to the migratory distance.

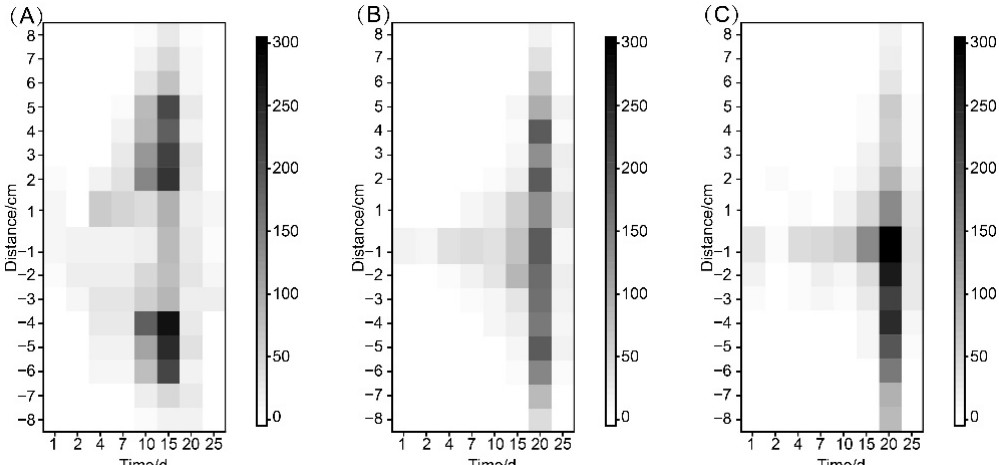

**Figure 3.** Distribution of *B. xylophilus* in 4-year-old *P. koraiensis* seedlings after inoculation (25 days). The shade of the square represents the number of nematodes in the different segments; a darker color corresponds to a higher number. The Y axis represents the distance between the inoculation point (1.5 cm) and sampling point. (**A**) Distribution of QH-1 in inoculated seedlings. (**B**) Distribution of NM-1 in inoculated seedlings. (**C**) Distribution of CM-1 in inoculated seedlings.

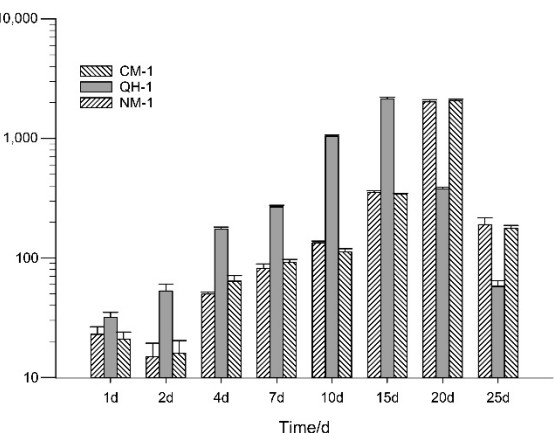

**Figure 4.** Number of *B. xylophilus* in 4-year-old *P. koraiensis* seedlings after inoculation (25 days). Error bars represent the mean ± SD of three replicates.

### 3.4. Internal Symptom Observations in Inoculated Trees

The internal symptoms of inoculated trees were observable using an optical microscope (Olympus, Japan). At 10 days after inoculation, the stem segments of the trees treated with sterile water (CK, CM-1, and NM-1) were healthy (Figure 5); however, half of the pith tissue and phloem was necrotic and had turned brown in the trees inoculated with QH-1. Moreover, the phloem and pith tissue were completely necrotic and brown in the trees inoculated with QH-1, while the stem segments of the CK-, CM-1-, and NM-1-treated trees were symptomless.

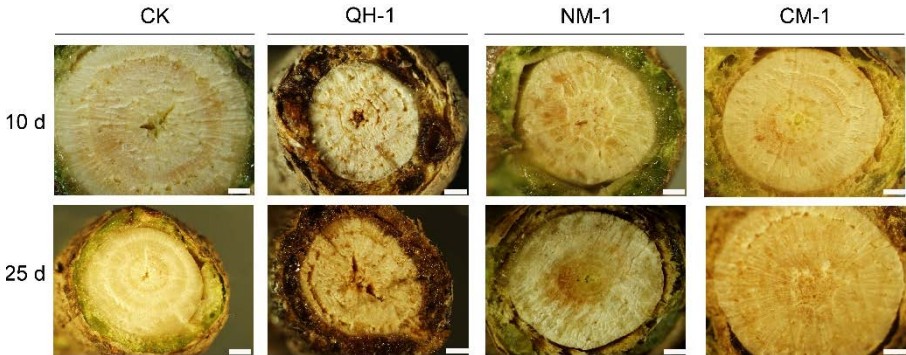

**Figure 5.** Cross section of *P. koraiensis* 1.5 cm below the point of inoculation with three different isolates of pinewood nematode. CK—seedlings inoculated with sterile water; QH-1, NM-1, and CM-1 denote the different isolate nematodes. Scale bar = 500 μm.

## 4. Discussion

This study showed that the three investigated isolates of *B. xylophilus* had varying rates of pathogenic effects on *P. koraiensis* seedlings. The virulence of QH-1 was significantly different from the other isolates. The results from the pathology tests showed that wilting symptoms began after 10 days and lasted up to 35 days in the inoculated seedlings, while the seedlings inoculated with QH-1 had the most rapid disease progression. These results are similar to those of a previous study on the infection of other pine seedlings with *B. xylophilus* [31–33]. The highly pathogenic PWN isolate caused the rapid onset and progression of disease in the inoculated seedlings.

The average number of nematodes recovered from the fungal culture after 2 weeks was different between the three isolates. The average number of NM-1 isolates was much greater than that of the QH-1 and CM-1 isolates. Interestingly, we found that the average number of nematodes recovered from the seedlings inoculated with QH-1 was greater than that of NM-1 and CM-1 after 35 days—indicating that the reproductive ability of

the nematodes from the fungal culture was different from that of the nematodes from the seedling culture—and the virulence of isolate QH-1 was much greater than the other isolates. We found that there was no relationship between the population of nematodes in a fungal culture and pathogenicity. This finding is similar to results reported in previous studies [25]. Some researchers suggest that the results of seedling inoculation experiments should be confirmed in the field and that the *B. cinerea* multiplication method is not reliable for verifying the pathogenicity of different isolates [34]. According to the results of this study, the high reproductive ability of *B. xylophilus* on fungal culture has no significant positive relationship with the pathogenicity of nematodes from different isolates, while the multiplication ability of *B. xylophilus* on seedling culture has a positive relationship with the pathogenicity of different nematodes.

The high mortality rate of inoculated seedlings directly reflects the high pathogenicity of the PWN. In this study, the mortality rate varied from 40% to 100%, and the QH-1 isolate caused the highest mortality of inoculated seedlings during the experimental period. However, the low mortality of the inoculated seedlings does not mean that the pathogenicity of the PWN was lower. Other factors, such as reproductive ability and distribution, may also influence the pathogenicity of PWN [24,35]. Further studies are needed to investigate which factors influence the pathogenicity of nematodes.

The early distribution of *B. xylophilus* in seedling culture is considered an important factor for PWNs. In our study, we found that, in the early stages of PWN migration, the migration ability of the QH-1 isolate was higher than that of CM-1 and NM-1. Kawaguchi (2006) [36] found that PWN migration in pine shoots infected by cortical resin canals was one indicator that could be used for verifying the resistance level of pine seedlings. Previous studies have shown that PWNs can migrate rapidly in susceptible pine trees. On the contrary, PWNs cannot migrate effectively in resistant pine trees [23,37]. Our study showed that the migration of the QH-1 isolate was more rapid than that of CM-1 and NM-1. We inferred that *P. koraiensis* is more susceptible to QH-1 than CM-1 and NM-1. According to the early symptoms of the inoculated trees and the internal symptoms of the stem cuttings, the QH-1 isolate caused pine wilt and pith tissue necrosis faster than the other isolates. We believe that the rapid migration of the QH-1 isolate hastened the progress of PWD.

Other factors can also affect the pathogenicity of the PWN in pine trees (e.g., environmental temperature, resistance of inoculated trees, etc.). Here, we suggest that the reproductive ability of the PWN in seedling culture, the mortality rate of inoculated seedlings, and PWN migration verify the virulence levels of the QH-1, CM-1, and NM-1 isolates. The most virulent isolate of *B. xylophilus* was QH-1. Early symptoms can also be used to differentiate virulence; however, that is a qualitative rather than quantitative analysis. The relationships between the different factors still require more investigation.

The first report of pine wilt disease in *P. koraiensis* was in Korea in 2006 [38]; a 95% mortality rate was observed from a pathogenicity test on 15-year-old *P. koraiensis* trees with 15,000 PWNs per tree. Other researchers also confirmed that PWNs caused high mortality in *P. koraiensis* [39]. In our study, *P. koraiensis* was susceptible to PWNs; however, the different PWN isolates caused different patterns of PWN development, showing varying migration and reproductive abilities in *P. koraiensis* seedlings. This indicates that *P. koraiensis* has a high degree of tolerance—but likely not resistance—to the PWN. We suggest that germplasm resource nurseries should be developed for *P. koraiensis* seedlings resistant to PWN, from which we can screen for and utilize resistant families of *P. koraiensis* to control PWD. Further investigation of the mechanisms of PWN resistance in *P. koraiensis* is needed in order to control PWD in the future.

## 5. Conclusions

All three *B. xylophilus* isolates had pathogenic effects on *P. koraiensis* seedlings. The reproductive and migration abilities of QH-1 were significantly higher than those of CM-1 and NM-1 in the seedling culture, and the mortality of the seedlings inoculated with QH-1

was greater than the other treatments. According to the onset of disease and mortality, the QH-1 isolate was the most virulent in *P. koraiensis*. These results suggest that *B. xylophilus* isolates with higher virulence have greater reproductive and migration abilities in *P. koraiensis* seedlings. The mortality of inoculated seedlings, migration ability, and reproductive ability (in trees) can differ in the pathogenicity between different isolates. The results of this study may support the investigation into screening resistant families of *P. koraiensis* to control PWD in order to decrease damage in the north of China.

**Author Contributions:** Y.-F.C., L.-F.W. and X.-Z.W. designed the study; Y.-F.C. and X.W. performed the experiments and analyzed the results; X.W. and M.X. assisted with the research; L.-F.W. and Y.-F.C. wrote the manuscript. All authors have read and agreed to the published version of the manuscript.

**Funding:** This work was supported by the National Key Research and Development Project of China (2021YFD1400905).

**Institutional Review Board Statement:** Not applicable.

**Informed Consent Statement:** Not applicable.

**Data Availability Statement:** The data presented in this study are available on request from the corresponding author. The data are not publicly available due to privacy restrictions.

**Conflicts of Interest:** The authors declare no conflict of interest.

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
