# Peer review of "Pathogenicity of Three Bursaphelenchus xylophilus (Steiner & Buhrer) Nickle. Isolates in Pinus koraiensis (Siebold & Zucc.) Seedlings"

_forests, doi:10.3390/f13081197_

Round 1
Reviewer 1 Report
The manuscript I have read is well written and able to publication in Forests (MDPI).
Some relevant references could be include for better sufficient background. For example:
https://www.researchgate.net/publication/279978648_The_most_simple_techniques_for_detection_and_laboratory_cultivation_of_woody_plant_wilt_nematodes_in_Russian_with_English_summary_SAMYE_PROSTYE_METODY_OBNARUZENIA_STVOLOVYH_NEMATOD_I_IH_LABORATORNOGO_.
In general the manuscript “Pathogenicity of three Bursaphelenchus xylophilus isolates in 2 Pinus koraiensis seedlings” can be published in Forests (MDPI) after proves that I asked above.
Author Response
Please write down "Please see the attachment.

Reviewer 2 Report
The manuscript entitled Pathogenicity of three Bursaphelenchus xylophilus isolates in Pinus koraiensis seedlings brings some knowledge to our science. However, the manuscript is still immature to be published without major improvement.
1. Scientific writing and English must be improved
2. Details are lacking in almost all sections.
3. Abstract should be revised that it should stand alone
4. some references in the material and methods are needed to support the current work.
5. Full names of abbreviations should be written in figure legends, and figure legends should be more descriptive.
6. Discussion and Conclusion should be expanded.
Round 2
Reviewer 2 Report
The authors made significant improvement to the manuscript and i recommend it for publication in Forests.